# Vulnerabilities in child well-being among primary school children: a cross-sectional study in Bradford, UK

Kate E Pickett ![ORCID] ,[1] Mildred Ajebon,[1] Bo Hou ![ORCID] ,[2] Brian Kelly ![ORCID] ,[2] Philippa K Bird,[3] Josie Dickerson ![ORCID] ,[2] Katy Shire ![ORCID] ,[2] Claire McIvor,[2] Mark Mon-Williams,[4] Neil Small,[5] Rosemary McEachan,[2] John Wright ![ORCID] ,[2] Deborah Lawlor[6]

[1]Health Sciences, University of York, York, UK
[2]Bradford Institute for Health Research, Bradford, UK
[3]Leeds Teaching Hospitals NHS Trust, Leeds, UK
[4]Department of Psychology, University of Leeds, Leeds, UK
[5]Health Studies, University of Bradford, Bradford, UK
[6]Bristol Medical School, University of Bristol, Bristol, UK

**Correspondence to**
Dr Kate E Pickett;
kate.pickett@york.ac.uk

## ABSTRACT

**Objective** To describe the prevalence of factors related to well-being among primary school children in a deprived multiethnic community in the UK.

**Design and participants** Cross-sectional survey of 15641 children aged 7–10 years in Born in Bradford's Primary School Years study: whole-classroom samples in 89 Bradford primary schools between 2016 and 2019.

**Main outcome measures** Prevalence estimates by ethnicity (%, 95% CI) of single and multiple vulnerabilities in child well-being within and across four domains (Home, Family, Relationships; Material Resources; Friends and School; Subjective Well-being).

**Results** Only 10% of children had no vulnerabilities in any domain of well-being; 10% had one or more vulnerabilities in all four domains. The highest prevalence estimates were for being bullied some or all of the time (52.7%, 95% CI: 51.9% to 53.4%), keeping worries to oneself (31.2%, 95% CI: 30.5% to 31.9%), having no park near home (30.8%, 95% CI: 30.1% to 31.5%) and worrying all the time about how much money their family has (26.3%, 95% CI: 25.6% to 27%). Boys were consistently significantly more likely than girls to report all of the vulnerabilities in the Home, Family and Family Relationships domain, and the majority of indicators in the other domains, and in all domains except Friends and School, boys were significantly more likely to have at least one vulnerability. Girls were significantly more likely to report not having many friends (16.7%, 95% CI: 15.9% to 17.6% vs 12.5%, 95% CI: 11.8% to 13.2%), being bullied some or all of the time (55.8%, 95% CI: 54.7% to 56.9% vs 49.7%, 95% CI: 48.6% to 50.8%) and feeling left out all the time (12.1%, 95% CI: 11.4% to 12.8%) versus (10.3%, 95% CI: 9.7% to 11.0%). Variations in vulnerabilities by ethnicity were complex, with children in black, Asian and minority ethnic groups sometimes reporting more vulnerabilities and sometimes fewer than white British children. For example, compared with children of Pakistani heritage, white British children were more likely to say that their family never gets along well (6.3%, 95% CI: 5.6% to 7.1% vs 4.1%, 95% CI: 3.6% to 4.6%) and to have no access to the internet at home (22.3%, 95% CI: 21% to 23.6% vs 18%, 95% CI: 17% to 18.9%). Children with Pakistani heritage were more likely than white British children to say they had no park near their home where they can play with friends (32.7%, 95% CI: 31.6% to 33.9% vs 29.9%, 95% CI: 28.6% to 31.3%), to report not having three meals a day (17.9%, 95% CI:

16.9% to 18.8% vs 11.9%, 95% CI: 10.9% to 12.9%) and to worry all the time about how much money their families have (29.3%, 95% CI: 28.2% to 30.3%) vs (21.6%, 95% CI: 20.4% to 22.9%). Gypsy/Irish Traveller children were less likely than white British children to say they were bullied some or all of the time (42.2%, 95% CI: 35.4% to 49.4% vs 53.8%, 95% CI: 52.3% to 55.3%), but more likely to say they were mean to others all the time (9.9%, 95% CI: 6.3% to 15.2% vs 4%, 95% CI: 3.5% to 4.7%) and can never work out what to do when things are hard (15.2%, 95% CI: 10.6% to 21.2% vs 9%, 95% CI: 8.2% to 9.9%). We considered six vulnerabilities to be of particular concern during the COVID-19 pandemic and associated national and local lockdowns: family never gets along well together; no garden where child can play; no nearby park where they can play; not having three meals a day; no internet at home; worried about money all the time. Pre-pandemic, 37.4% (95% CI: 36.6% to 38.3%) of Bradford children had one of these vulnerabilities and a further 29.6% (95% CI: 28.9% to 30.4%) had more than one.

**Conclusions** Although most primary school children aged 7–10 in our study had good levels of well-being on most indicators across multiple domains, fewer than 10% had no vulnerabilities at all, a worrying 10% had at least one vulnerability in all the four domains we studied and two-thirds had vulnerabilities of particular concern during the COVID-19 lockdowns.

---

## STRENGTHS AND LIMITATIONS OF THIS STUDY

⇒ A uniquely large study of contemporary child well-being in the UK with a high-response rate and including a broad group of ethnicities, generalisable to other UK cities with high levels of ethnic diversity and deprivation.

⇒ Subjective measures of a wide range of experiences capture how children perceive their lives.

⇒ A descriptive study of prevalence of vulnerabilities in well-being, unable to examine causes.

## INTRODUCTION

In 2007, the UK ranked bottom of 21 countries in UNICEF's first report on child well-being in rich countries,[1] and from 2009 through 2019 the Good Childhood inquiry[2]

and subsequent reports from the Children's Society[3] found increasing levels of unhappiness and mental distress among children and young people in the UK. The State of the Nation 2019 report on children and young people's well-being suggested a need to 'understand well-being across different groups' and to 'use a range of measures to understand their experience'.[4]

UK research suggests that well-being declines as children and young people grow older[4] but little is known about well-being in primary school aged children in the UK as previous studies, including the Good Childhood surveys, Understanding Society and HeadStart, included children over 10 years old only. The impact of ethnicity on child well-being at any age is unclear and most previous UK studies have been unable to examine well-being across a wide range of ethnicities.[4]

The global COVID-19 pandemic and restrictions, such as lockdowns that closed schools, heightened concerns about child well-being across the world. The prevalence of pre-existing child vulnerabilities and risk factors that could potentially require mitigation and targeted intervention during and after the pandemic has been a concern, particularly in poor, ethnically diverse urban settings.[5]

To provide evidence to meet these knowledge gaps, we examined child well-being among primary school children, aged 7–10 years, in a city in the North of England. Bradford is the fifth largest metropolitan district in England with one of the youngest and most ethnically diverse populations.[6] Bradford also has some of the highest levels of poverty and ill-health in England; almost a quarter of Bradford children live in poverty and 24% are obese at age 10/11.[7] The aim of this paper is to describe the prevalence of vulnerabilities in child well-being by ethnicity and sex in a deprived community. While this research was undertaken in the city of Bradford and has been used to support local policy during the COVID-19 pandemic,[8] we believe the results have wider relevance for deprived multiethnic urban settings.

## METHODS
### Setting and participants
Born in Bradford (BiB) is a cohort study of 13 500 children born between 2007 and 2011.[9 10] Between 2016 and 2019, BiB administered the Primary School Years study as a cross-sectional survey of well-being of children aged 7–10 years in 89 Bradford schools. We surveyed whole classrooms, including children previously recruited to the ongoing BiB cohort study as well as non-BiB children. The study protocol and detailed methods of school recruitment are described elsewhere[11 12] and the STROBE (Strengthening the Reporting of Observational Studies in Epidemiology) statement can be found in online supplemental material 1. Schools were given information sheets and opt-out consent forms to give to parents of children in eligible year groups ahead of the school visit; one school asked for opt-in consent to be used and this

was accommodated. Fewer than 5% of parents opted out of the study.

### Design and procedures
The well-being survey ('Me and My Life') was developed for this study (see online supplemental material 2), drawing on questions asked of children in the Millennium Cohort Study,[13] the International Survey of Children's Well-being[14] and the Avon Longitudinal Study of Parents and Children.[15] The survey aimed to assess children's well-being in multiple domains, including: happiness, health, material well-being, relationships with family and friends, school experience, neighbourhood, aspirations and acculturation. 'Think aloud' testing with a small group of children was used to check understanding and face validity. During administration, whole classrooms completed the survey at the same time, supported by three research facilitators.

Schools provided class lists, including date of birth, sex and ethnicity. Child ethnicity is reported by parents on registration with a school. The ethnicity information provided from school records contained 192 categories. These were re-coded into 18 broad groups in line with the 2011 census categorisation of ethnic groups in the UK and for the analyses presented here collapsed into 10 categories: Pakistani, Bangladeshi, Indian, black/black British, white British, mixed, Gypsy/Irish Traveller, other white, other and unknown. Where ethnicity was missing it was possible to supplement, for BiB children only, with data held by BiB, as BiB requests regular data updates of child data from the local authority (n=311).

To assess vulnerabilities in well-being, we grouped questions within four domains: (1) Home, Family and Family relationships, (2) Material Resources, (3) Friends and School, (4) Subjective (self-reported) Well-being. Within each domain, we defined 'vulnerabilities' on (a) the basis of research literature showing these to be well-established childhood risk factors for long-term health, well-being, educational attainment and social mobility and (b) subsequent consultation with the vulnerabilities workstream of the Bradford Institute for Health Research COVID-19 Scientific Advisory Group (https://www.bradfordresearch.nhs.uk/vulnerable-groups/). The risk factors categorised as vulnerabilities include, by domain:

► Home, Family, Family relationships: family never gets along well together, no garden where child can play, no nearby park where they can play, never plays in park.
► Material Resources: no warm winter coat, not having three meals a day, no internet at home, worries about money all the time.
► Friends and School: does not like school, not many friends, bullied some or all of the time, mean to others all the time, feel left out all the time.
► Subjective Well-being: never happy, always sad, ill/unwell all of the time, keeps worries to self, can never work out what to do when things are hard.

**Recruited N=17,774**

BiB N=6,882
Non-BiB=10,892
Primary schools N=89

**Included N=15,641**

BiB N=6,147
Non-BiB=9,494
Primary schools N=89

**Excluded N=2,133**

BiB N=735
Non-BiB=1,398
Primary schools N=0

Reasons included school absence, refusal and
insufficient time to test

**Figure 1** Recruitment of schools, Born in Bradford cohort members and non-Born in Bradford children to the BiB primary school years well-being survey. BiB, Born in Bradford.

### Data analysis

Socio-demographics and the prevalence of vulnerabilities in child well-being are presented as counts and/or percentages with 95% CIs, for all children and by sex and ethnicity. We also describe the number and per cent of children with multiple vulnerabilities within and across domains of well-being by sex and ethnicity. Children with missing data for any answer are excluded from the relevant analyses and number of missing reported in figures and tables.

### RESULTS

Figure 1 shows the number of schools and children included in the study.

Table 1 shows the socio-demographics of the 15 641 survey respondents. Figure 2 presents the prevalence of each vulnerability across all domains for all children (see also online supplemental table S1); figure 3 presents the percentage of all children who have zero, one and more than one vulnerability within each domain (see also online supplemental table S2).

Although the prevalence of vulnerability varied by item and domain, within each domain there is an item with notably high prevalence, with more than one in four children reporting vulnerability: no park near home (30.8%, 95% CI: 30.1% to 31.5%); worries about money all the time (26.3%, 95% CI: 25.6% to 27.0%); bullied some or all of the time (52.7%, 95% CI: 51.9% to 53.4%); and keeping worries to self (31.2%, 95% CI: 30.5% to 31.9%).

We considered six vulnerabilities in child well-being would be of particular concern during lockdowns and school closures associated with the COVID-19 pandemic: family never gets along well together; no garden where child can play; no nearby park where they can play; not having three meals a day; no internet at home; worried about money all the time. Before the pandemic 37.4% (36.6%–38.3%) of children had one of these vulnerabilities and a further 29.6% (28.9%–30.4%) had more than one.

Although the majority of children had no vulnerabilities in the Home, Family, Relationships domain (55.9%, 95% CI: 55.0% to 56.7%) and the Subjective Well-being domain (57.4%, 95% CI: 56.6% to 58.2%), the majority of children had one or more vulnerabilities in the Material Resources and Friends and School domains. Only 10% (add 95% CI) of children had no vulnerabilities in any domain of well-being. Figure 4 is a Venn diagram that shows how vulnerabilities overlap across domains. One thousand four hundred and ninety-four (10%) children had no vulnerabilities. Each oval in the Venn diagram includes all the children who had one or more vulnerabilities within that domain; so, for example, 2% of children had vulnerabilities in Home, Family, Relationships *and* Subjective Well-being but not in other domains. One thousand five hundred and nineteen children (10%) had one or more vulnerabilities in all four domains.

Table 1   Socio-demographic characteristics of 15 641 Bradford primary school children

|  | Number | Per cent |
|---|---|---|
| **Sex** | | |
| Female | 7647 | 48.90 |
| Male | 7994 | 51.10 |
| **Age (years)** | | |
| 6 | 2 | <0.01 |
| 7 | 5264 | 33.70 |
| 8 | 6901 | 44.10 |
| 9 | 3175 | 20.30 |
| 10 | 293 | 1.90 |
| 11 | 1 | <0.01 |
| 12 | 3 | <0.01 |
| 13 | 2 | <0.01 |
| Missing | | |
| **Ethnicity** | | |
| Pakistani | 7031 | 45.00 |
| Bangladeshi | 471 | 3.00 |
| Indian | 357 | 2.30 |
| Black/black British | 277 | 1.80 |
| White British | 4247 | 27.10 |
| Mixed | 900 | 5.70 |
| Gypsy/Irish Traveller | 190 | 1.20 |
| White other | 707 | 4.50 |
| Other | 425 | 2.70 |
| Unknown | 1036 | 6.60 |

## Variation by sex

Boys were consistently significantly more likely to report all of the vulnerabilities in the Home, Family and Family Relationships domain, although the sex differences were not large (see online supplemental table S3 for full results, including 95% CIs, for all domains by sex). There were no sex differences in having access to the internet at home, but boys were significantly more likely to report not having a winter coat (11.1%, 95% CI: 10.4% to 11.9% vs 8.9%, 95% CI: 8.2% to 9.6%), not having three meals a day (17.7%, 95% CI: 16.8% to 18.6% vs 14.0%, 95% CI: 13.3% to 14.9%) and worrying about money all the time (28.9%, 95% CI: 28.0% to 30.0%) versus (23.4%, 95% CI: 22.5% to 24.4%). Within the Friends and School domain, boys were significantly more likely to not like school (18.0%, 95% CI: 17.1% to 18.8% vs 8%, 95% CI: 7.4% to 8.6%) and to be mean to others all the time (7.0%, 95% CI: 6.4% to 7.6% vs 3.4%, 95% CI: 3.0% to 3.8%), while girls were significantly more likely to report not having many friends (16.7%, 95% CI: 15.9% to 17.6% vs 12.5%, 95% CI: 11.8% to 13.2%), being bullied some or all of the time (55.8%, 95% CI: 54.7% to 56.9% vs 49.7%, 95% CI: 48.6% to 50.8%) and feeling left out all the time (12.1%, 95% CI: 11.4% to 12.8% vs 10.3%, 95% CI: 9.7% to 11.0%). Within the Subjective Well-being domain there were no sex differences in being always sad or always ill or unwell, but boys were significantly likely to report never being happy (5.2%, 95% CI: 4.7% to 5.7% vs 2.4%, 95% CI: 2.1% to 2.8%), keeping worries to themselves (33.7%, 95% CI: 32.7% to 34.8% vs 28.5%, 95% CI: 27.5% to 29.6%) and not being about to work out what to do when things are hard (9.6%, 95% CI: 9.0% to 10.3% vs 7.5%, 95% CI: 6.9% to 8.1%). In all domains except for Friends and School, boys were significantly more likely to have at least one vulnerability, compared with girls.

## Variation by ethnicity

Figures 5–8 shows the percentages (and 95% CIs) of primary school children with at least one vulnerability within each of the domains of child well-being, by

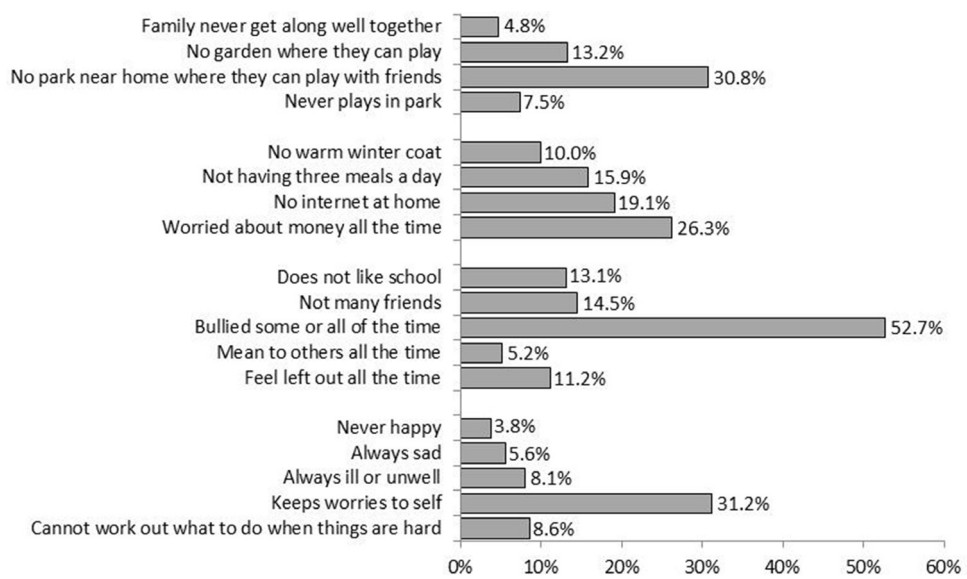

**Figure 2**   Per cent of primary school children with each vulnerability across domains of well-being.

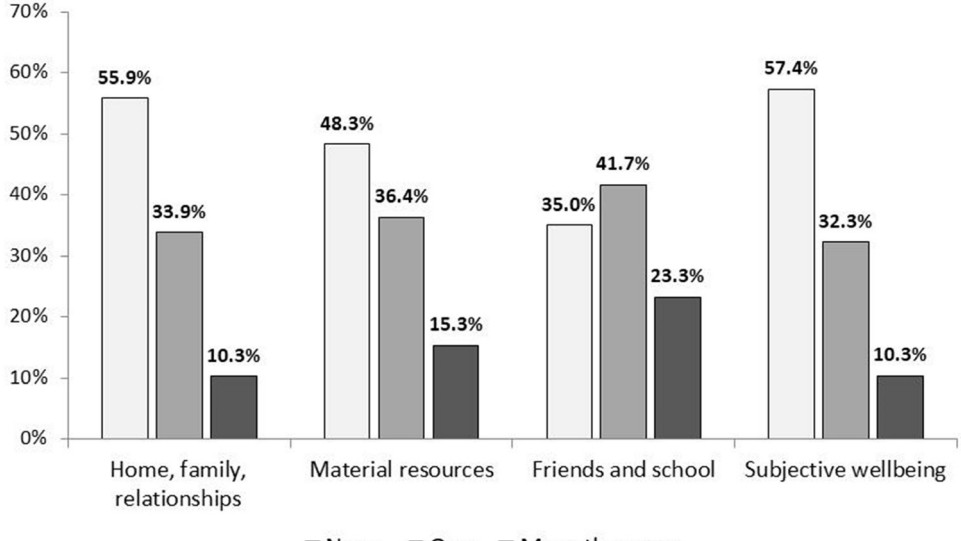

**Figure 3** Per cent of primary school children with zero, one or more than vulnerability in each domain of well-being.

ethnicity. Full results are given in online supplemental table S4A–D.

### Home, Family, Relationships

Compared with white British children, Pakistani, white other and black/black British children were significantly less likely to say their family never get along well together. White other and other children were significantly more likely to not have a garden where they can play, compared with white British children. Compared with white British children, Pakistani children were significantly more likely and black/black British children significantly less likely to have no park near home where they can play with friends. Compared with white British children, Pakistani children were significantly less likely to say they never play in a park (figure 5).

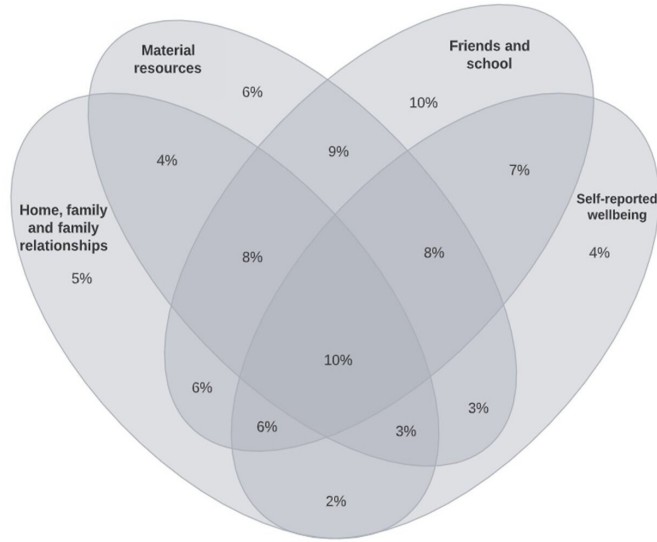

10%: no vulnerabilities

**Figure 4** Per cent of children with vulnerabilities within and across domains.

### Material Resources

There were no significant ethnic differences in the proportion of children who did not have a warm winter coat. Compared with white British children, Pakistani, Indian and Gypsy/Traveller children, as well as children of mixed ethnicity were significantly more likely to report not having three meals a day. Pakistani, white Other, Indian and black/black British children were significantly less likely to report having no internet at home than white British children (figure 6).

Compared with white British children, Pakistani, white other, Bangladeshi and Gypsy/Irish Traveller children were significantly more likely to worry all the time about how much money their families have.

### Friends and School

Pakistani, white other and Indian children were significantly less likely to report not liking school than white British children, and Bangladeshi and black/British were significantly less likely to report not having many friends. Compared with white British children, white other and Gypsy/Irish Traveller children were significantly less likely to say they were bullied some or all of the time, and Pakistani children and Gypsy/Irish Traveller children were significantly more likely to say they were mean to others all the time. There were no significant ethnic differences in the proportion of children who felt left out by others all the time (figure 7).

### Subjective Well-being

There were no significant ethnic differences in the proportion of children who said they were never happy or were always sad or who said they keep worries to themselves. Compared with white British children, only Pakistani children were significantly more likely to say they were always ill or unwell. Compared with white British children, Bangladeshi children were significantly less likely and Gypsy/Irish Traveller children more likely, to

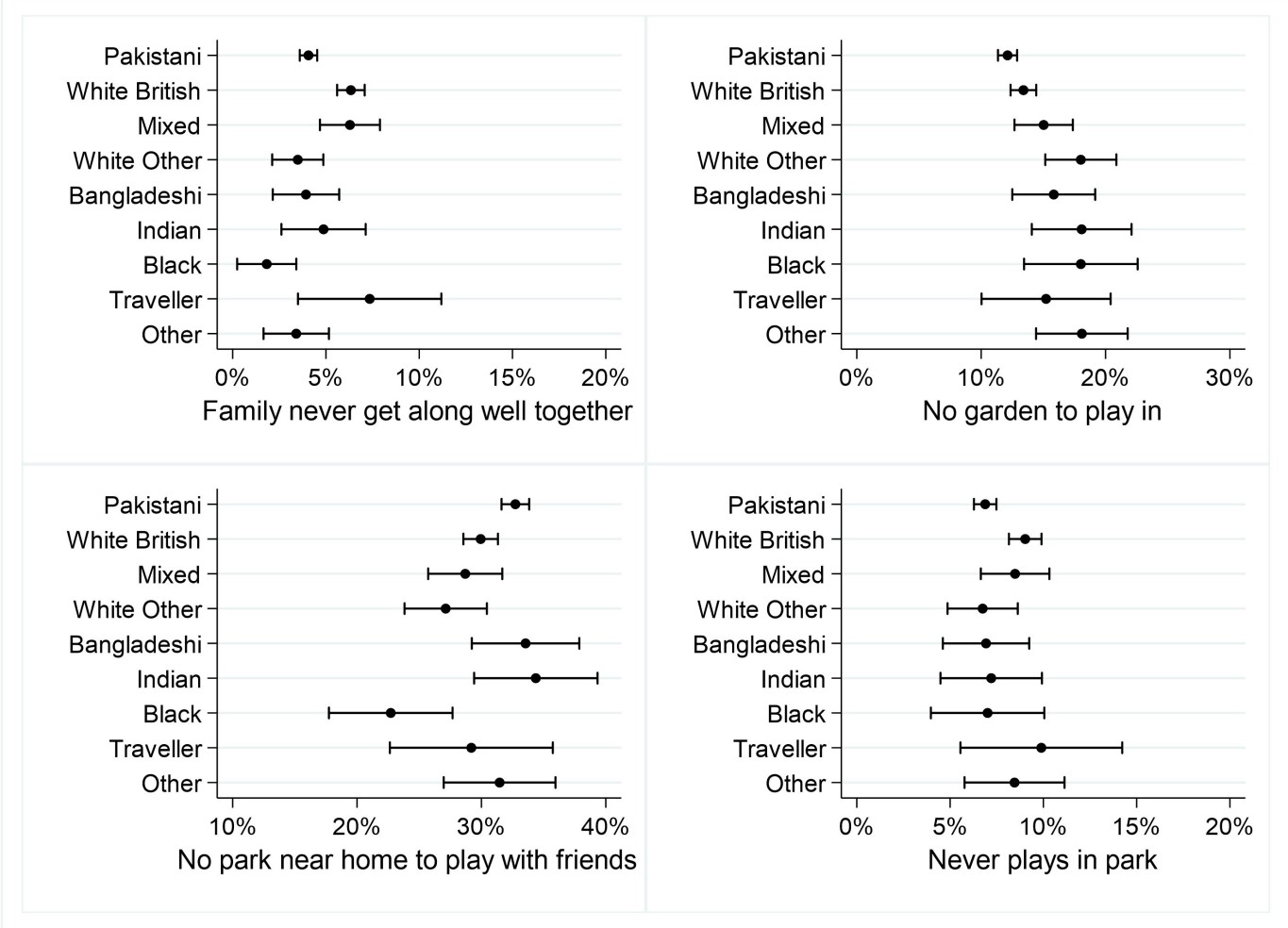

**Figure 5** Per cent of primary school children with at least one vulnerability in the Home, Family and Family Relationships domain.

say they cannot work out what to do when things are hard (figure 8).

## DISCUSSION
### Statement of principal findings
Although most primary school children aged 7–10 in Bradford had good levels of well-being on most indicators across multiple domains, fewer than 10% had no vulnerabilities at all, and a worrying 10% had more than one vulnerability in all the four domains we studied. Variations in vulnerabilities by ethnicity were complex, with children in black, Asian and minority ethnic groups sometimes reporting more vulnerabilities and sometimes fewer than white British children. We found that boys reported significantly more vulnerabilities for all indicators in the Home, Family and Family Relationships domain, as well as for the majority of indicators in the Material Resources, Friends and School, and Subjective Well-being domains. This contrasts with the finding of few consistent sex differences in well-being and a trend towards worse well-being in girls in the UK State of the Nation 2019: children and young people's well-being report,[4] and the general picture in the research literature of worse well-being for adolescent girls[16 17] and inconsistent sex differences among adults.[18 19] These differences may be related to the younger age of our sample compared with many other studies which focus on secondary school pupils and adolescents. Future BiB data collection will enable analysis of whether these sex differences in well-being persist after children transition to secondary school. Our findings highlight a need for greater attention to boys' well-being among primary age children in both research and local practice. There is evidence of beneficial effects of male-targeted health promotion interventions to improve well-being among male adolescents,[20] however the evidence base on interventions to support well-being among primary school aged boys remains limited. This finding also suggests that worse well-being among adolescent girls might not be inevitable if prevention efforts were started before secondary education.

We found complex variations in vulnerabilities by ethnicity, with children in black, Asian and minority ethnic groups sometimes reporting more vulnerabilities and sometimes fewer than white British children. For

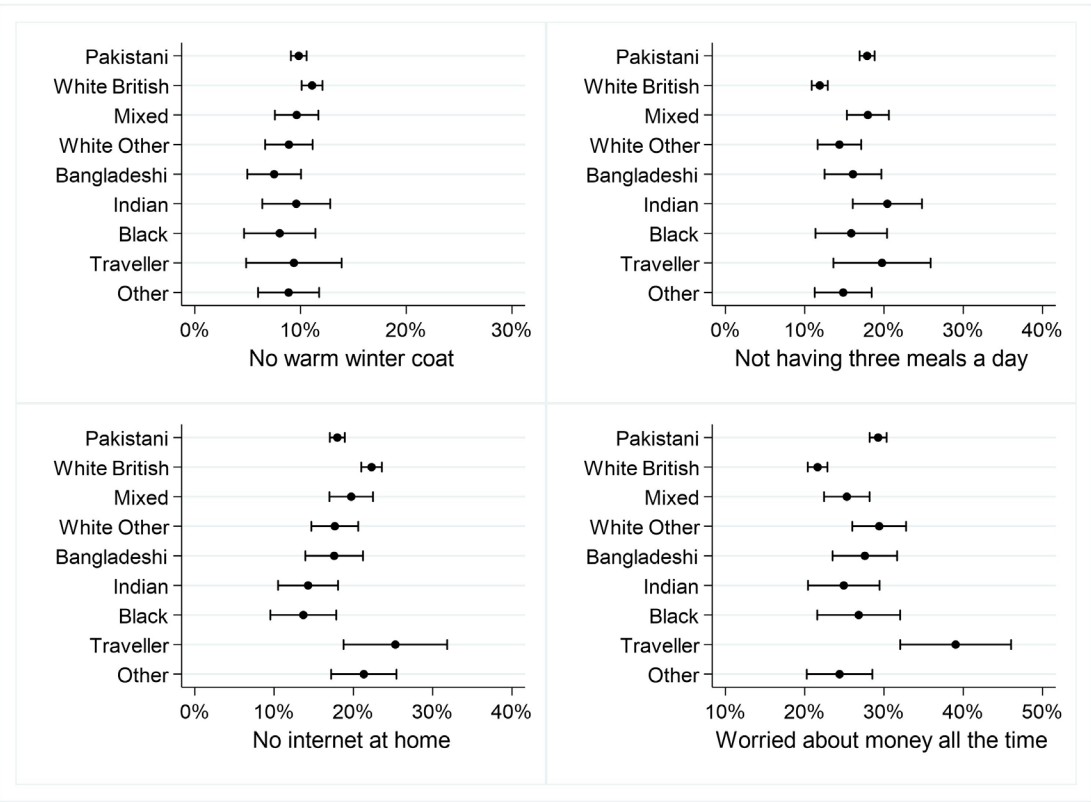

**Figure 6** Per cent of primary school children with at least one vulnerability in the Material Resources domain.

example, compared with children of Pakistani heritage, white British children were more likely to say that their family never gets along well and to have no access to the internet at home, whereas children with Pakistani heritage were more likely than white British children to say they had no park near their home where they can play with friends, to report not having three meals a day and to worry all the time about how much money their families have. The UK State of the Nation 2019: children and young people's well-being report drew attention to the importance of 'moving beyond the average' in understanding well-being but reported 'no discernible differences in well-being by…ethnicity' while recognising that small sample sizes might have obscured differences and reporting statistics only for five broadly defined ethnic groups.[4] We provide a detailed description of well-being for nine ethnic groups, so that nuances in ethnic variation can be better understood, including for smaller ethnic groups. For example, we found that Gypsy/Irish Traveller children were less likely than white British children to say they were bullied some or all of the time, but more likely to say they were mean to others, can never work out what to do when things are hard, worry all the time about how much money their family has, and not have three meals a day.

### Strengths and weaknesses of the study

To our knowledge, ours is the only large contemporary study of well-being in primary school aged children in England, with sufficient numbers of children belonging to some of the UK's smaller black, Asian and minority ethnic communities to be able to describe their well-being. Our approach to school recruitment, opt-out consent and whole classroom testing, along with BiB's longstanding trusting relationships with schools and communities ensured a very high-response rate. With coverage of three-quarters of primary schools and a response rate of 88% we do not believe that our findings are materially affected by selection or information bias with regards to ethnicity. In line with contemporary epidemiological good practice, we did not estimate multivariable models that included ethnicity as an independent variable to avoid difficulties inherent in interpreting the ethnicity coefficient in models controlling for other variables.[21] However, in line with current UK social science practice, we believe that the description of ethnic differences is appropriate in social policy oriented literature.[22] Although our study is based in a single city, it is likely to be generalisable to other UK cities with high levels of ethnic diversity and deprivation.

The fact that our measures of well-being are self-reported by children is both a strength and a potential weakness. It has been argued that, for adults and children alike, the subjective experience of quality of life matters, and that, as feelings have a demonstrable objective neuropsychological reality and are correlated with objective health, social and economic outcomes, they are worthy of measurement and inference.[23 24] Clearly, our

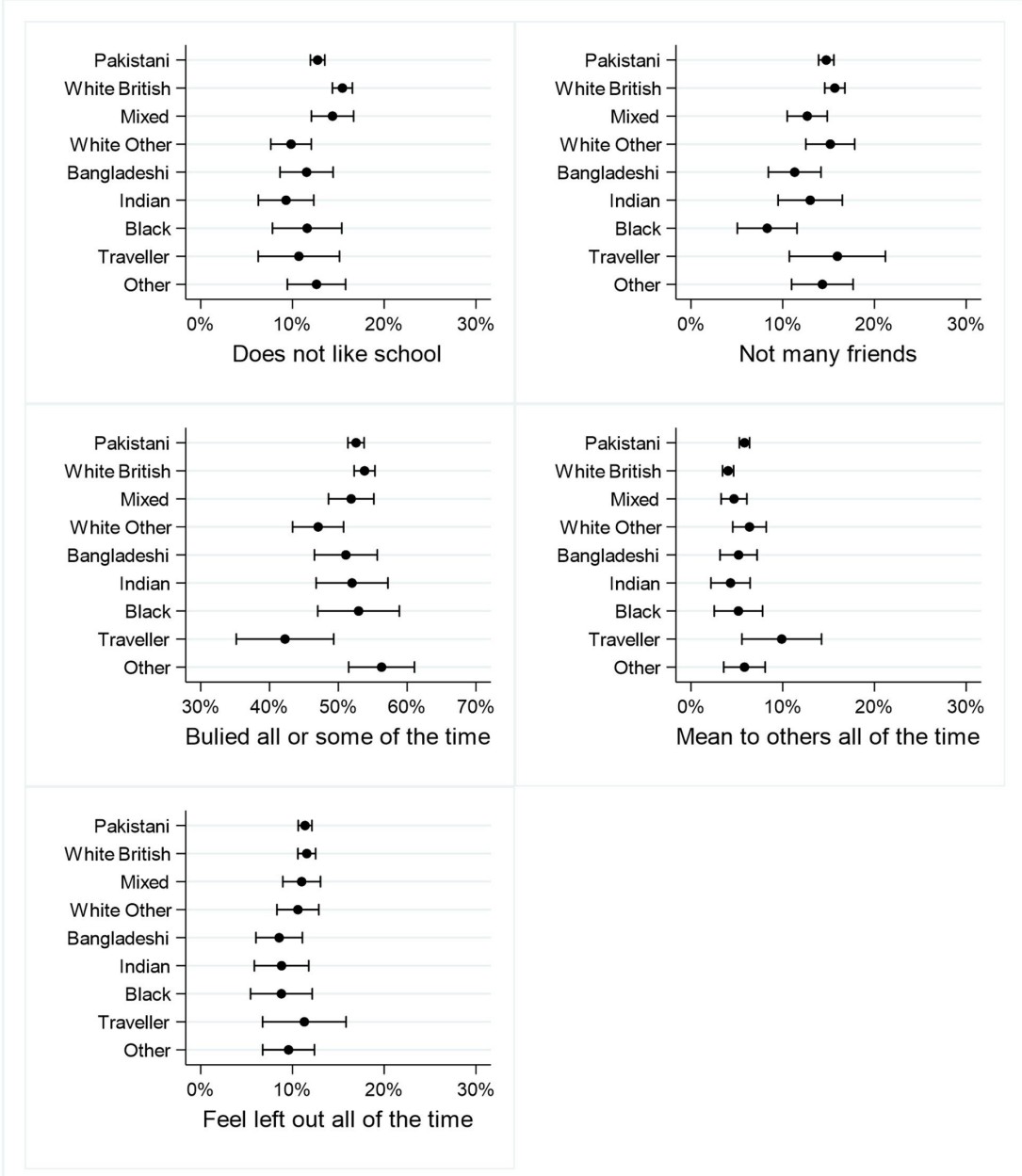

**Figure 7** Per cent of primary school children with at least one vulnerability in the Friends and School domain.

measures of vulnerability not only include things that can only be measured subjectively, such as happiness or feeling left out, but also things that arguably would be better measured objectively, for example, having access to a garden or three meals a day. However, it is possible that even for measures such as these, children's perceptions that they lack material resources may be important even if they actually have the resources.

Our data collection objectives and approach to categorising vulnerabilities in well-being was to capture a broad range of children's circumstances and experiences that might underpin variation in developing life trajectories. This is distinct from the adverse childhood experiences approach[25] and the definition and measurement of extreme child vulnerabilities collated and reported by the Children's Commissioner for England (https://www.

childrenscommissioner.gov.uk/chldrn/), both of which measure factors that put children at immediate, as well as long-term, risk of harm. Our objective was to collect data on more common well-being outcomes that are also risk factors with known long-term consequences for health and well-being, to better understand children's subjective experiences across the whole population and to measure well-being itself rather than social indicators, such as determinants of well-being, or well-becoming.[24] In our school-based whole classroom surveys, time constraints and study design prevented data collection and analyses that would have allowed us to go beyond description and investigate the causes of vulnerabilities in child well-being. For BiB children only, we examined associations between neighbourhood area deprivation (Index of Multiple Deprivation) and vulnerabilities. We found no

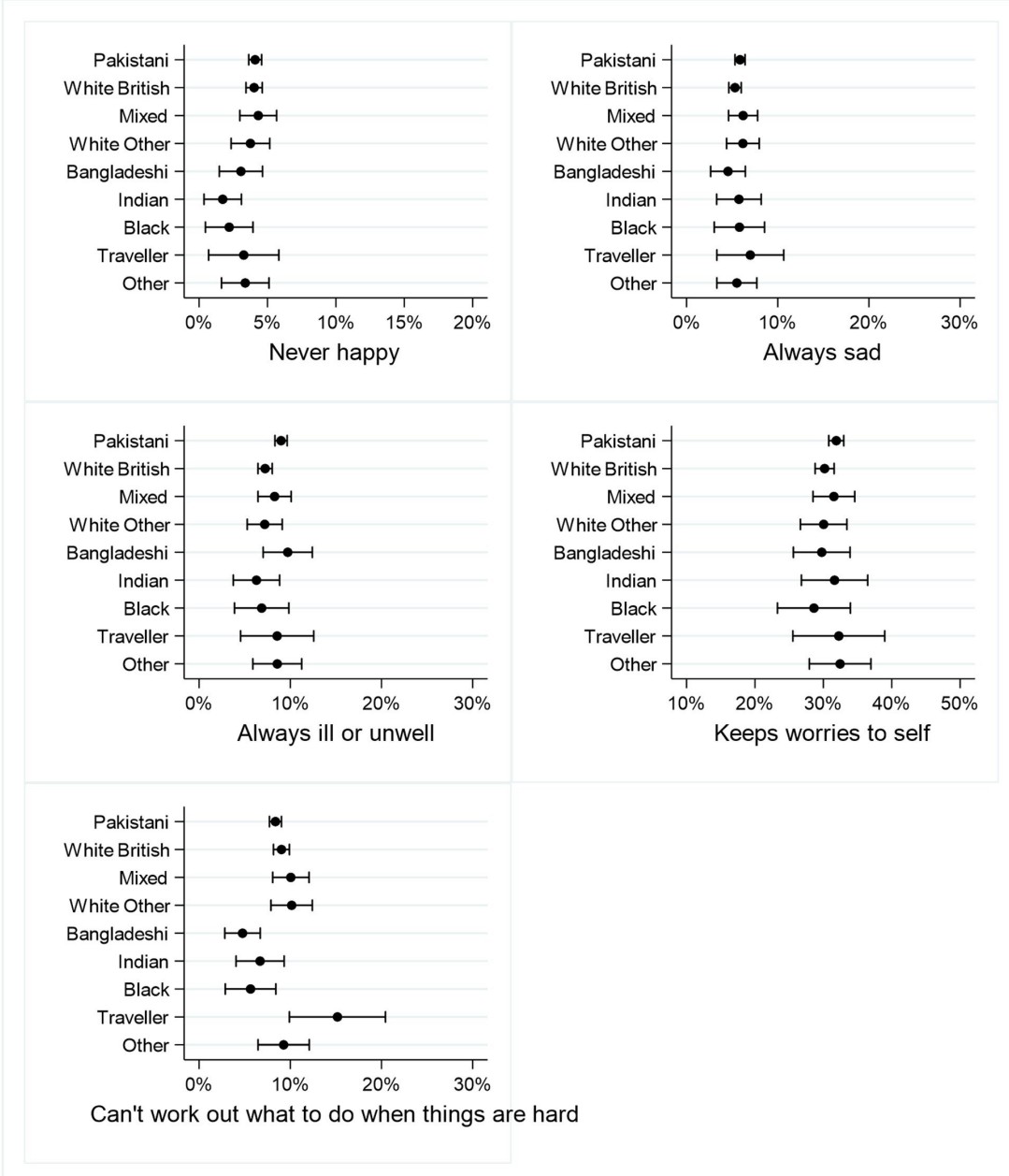

**Figure 8** Per cent of primary school children with at least one vulnerability in the Subjective Well-being domain.

significant associations for any domain except for more vulnerabilities in Material Resources at higher levels of deprivation, and as lack of material resources is definitional for deprivation do not report those analyses here (see online supplemental table S5). We plan future data collection with this sample to allow us to examine how vulnerabilities at primary school ages shape trajectories of well-being across time.

### Implications for providers of services to children and policymakers

### Implications for services and schools

Our findings showed the importance of friendships and school environments as key stressors for children. Schools are well placed to identify concerns and intervene early to

support children's well-being. There is a growing evidence base on 'what works' to support well-being and mental health in schools,[26] however most studies have focused on adolescent age groups.

Well-being is also increasingly recognised as a central aim and focus of education, for example, through the Well Schools Movement (https://www.well-school.org/), and the roll out of mental health support in schools.[27] As visible, trusted spaces in communities, schools can also be employed to engage and connect with families and enhance access to support where required.[28] Although all schools work hard to create safe and supportive environments for children, the high levels of bullying and social isolation that we report suggests that more needs to be done to make this a reality for all children, and to address

inequalities in well-being by ethnicity. This could be facilitated through better resourcing for schools to support well-being and create safe and supportive environments.

From a public health perspective, multisectoral approaches are required to integrate access to support across services, with schools at the centre. Pathways need to facilitate identification and provision of support across services and sectors covering the domains we have identified as key concerns for children's well-being—including from schools to local welfare services, voluntary sector mental health support, healthcare and early help. In the context of limited evidence on interventions to reduce inequalities in children's well-being by sex and ethnicity, local monitoring and evaluation needs to disaggregate data to understand how services can mitigate inequalities.

### Implications for national and local policymakers

Our research also highlights the importance of structural factors, from household finances and resources to green space. Action on the wider and structural determinants of well-being is critical to reduce inequalities in children's well-being, including the wide-ranging inequalities by ethnicity. Bradford is the 13th most deprived local authority in England (Indices of deprivation 2019), and many of the schools studied fall within the most deprived neighbourhoods in England. National policies to prevent children growing up in poverty and deprived neighbourhoods include uplifts to child benefit and abolishment of the two-child limit, improved access to free school meals and improvements to the National Living Wage.[29] There is a need to direct further resources to schools, especially in deprived areas, to facilitate work to support well-being and reduce inequalities. At a local level, local authorities, schools and other services can take action to ensure families can access advice services, limit the costs of the school day and school holidays and facilitate access to high quality green space.

Implications in the context of the COVID-19 pandemic: Our findings suggest that, prior to the COVID-19 pandemic, a substantial number of children were experiencing suboptimal well-being with potential lifelong consequences, with implications for all providers of services to children, as well as local and national policymakers. As the economic impact of the pandemic is increasing child poverty, income insecurity, food and housing insecurity, parental mental health challenges and family violence, the need for attention to the identification and amelioration of vulnerability is heightened.[30] The International Society for Social Pediatrics and Child Health has called for the adoption of 'a Child Rights Based Approach (CRBA) to respond to the COVID-19 pandemic, and to advance a future in which the health, development and well-being of children and youth are prioritised with explicit strategies to reduce health inequities and advance social justice' (https://www.issop.org/2020/06/01/issop-covid-19-declaration/). Such an approach can underpin and inform services and policymaking at all levels.

As the population was encouraged to 'lockdown' at home and schools closed for almost all children (and attendance was very low for those who were eligible to go to school) for varying periods, the response to the pandemic will have affected children's well-being (as well as their educational trajectories[31] differently, depending on their circumstances and context. Children who found school a nurturing refuge from difficult home situations will have suffered from school shutdowns, whereas others who found school a challenging environment may have been happy to be at home for a prolonged period; most children have probably experienced both positive and negative impacts.[32 33] In addition, our subsequent research has confirmed that the high prevalence of children with no access to the internet at home and no access to a garden or park has meant that many children have at times been unable to study[32] or be physically active.[34]

Locally, our research team has engaged with our local authority, public health, National Health Service and voluntary sector colleagues to identify principles to mitigate the impact of the pandemic on vulnerable groups (https://www.bradfordresearch.nhs.uk/wp-content/uploads/2020/04/CSAG-Briefing-Paper-Vulnerable-Groups-and-Recovery-230420.pdf). We have proposed additional support for children and young people in post-pandemic recovery planning, including assessing the impact of decision-making on children, prioritising children's needs, providing extra support for vulnerable children and monitoring impacts on children's lives and well-being.

### Conclusions and implications for public health

Among primary school children aged 7–10 in a deprived Northern city in the UK, fewer than 10% had no vulnerabilities in Home, Family and Family Relationships, Material Resources, Friends and School and Subjective Well-being, while 10% had more than one vulnerability in all of these domains. These children are at risk of poor health, social and economic trajectories throughout their lives. For public health, our findings highlight the emergence of inequalities in child well-being at an early age, and the need for early intervention to improve children's lives and reduce impacts later in life. Multisectoral approaches are required to address the wide range of domains, across health, schools and wider local services, with integrated pathways to identify and support children. Inequalities in child well-being could be mitigated by national policy and local practice focused at structural factors, such as poverty eradication and school-based approaches to identify and support well-being using both universal approaches and targeted support for the most vulnerable.

**Acknowledgements** This study was funded by a joint grant from the UK Medical Research Council and UK Economic and Social Science Research Council (MR/N024397/1), with support from a Wellcome Trust infrastructure grant (WT101597MA) and the National Institute for Health Research under its Applied Research Collaboration Yorkshire and Humber (NIHR200166). The funders had no role in the design of the study, the collection, analysis or interpretation of the

data; the writing of the manuscript, or the decision to submit the manuscript for publication. All authors are independent of the funders, had full access to all of the data (including statistical reports and tables) in the study and can take responsibility for the integrity of the data and the accuracy of the data analysis. Views expressed in this paper are those of the authors and not necessarily those of any funder. Born in Bradford is only possible because of the enthusiasm and commitment of the Children and Parents in BiB. We are grateful to all the participants, health professionals, schools and researchers who have made Born in Bradford happen. Research results are disseminated through https://borninbradford.nhs.uk/our-findings/ and actively through various engagement activities to parents, children and young people and stakeholders.

**Contributors** KP, DL, JW, RM, MM-W and NS obtained funding for the study, and with PKB and JD conceived the research questions and analyses. BK, BH, MA and KS cleaned and curated the data, conducted analyses and contributed to the methodology. KP is guarantor and drafted the first manuscript. KP, PKB and CM revised the manuscript. All authors read and approved the final version of the manuscript.

**Funding** This report is independent research funded by the Medical Research Council, the Wellcome Trust and the National Institute for Health Research Yorkshire and Humber ARC. The views expressed in this publication are those of the author(s) and not necessarily those of the National Institute for Health Research or the Department of Health and Social Care.

**Competing interests** None declared.

**Patient and public involvement** Patients and/or the public were involved in the design, or conduct, or reporting, or dissemination plans of this research. Refer to the Methods section for further details.

**Patient consent for publication** Not applicable.

**Ethics approval** Ethical approval has been obtained from the National Health Service Health Research Authority Yorkshire and the Humber (Bradford Leeds) Research Ethics Committee (reference: 16/YH/0062). Parents gave opt-out consent.

**Provenance and peer review** Not commissioned; externally peer reviewed.

**Data availability statement** Data are available upon reasonable request. Scientists are encouraged and able to use BiB data, which are available through a system of managed open access (borninbradford@bthft.nhs.uk).

**ORCID iDs**
Kate E Pickett http://orcid.org/0000-0002-8066-8507
Bo Hou http://orcid.org/0000-0001-5337-6560
Brian Kelly http://orcid.org/0000-0003-1834-2992
Josie Dickerson http://orcid.org/0000-0003-0121-3406
Katy Shire http://orcid.org/0000-0002-2093-181X
John Wright http://orcid.org/0000-0001-9572-7293

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
