## [Reviewer comments · BMJ Open]

ARTICLE DETAILS

TITLE (PROVISIONAL)	Vulnerabilities in child wellbeing among primary school children: a cross-sectional study in Bradford, UK
AUTHORS	Pickett, Kate; Ajebon, Mildred; Hou, Bo; Kelly, Brian; Bird, Philippa K; Dickerson, Josie; Shire, Katy; Mclvor, Claire; Mon-Williams, Mark; Small, Neil; McEachan, Rosemary; Wright, John; Lawlor, Deborah

VERSION 1 – REVIEW

REVIEWER	Ward, Aimee Kent State University, Geography
REVIEW RETURNED	31-May-2021

GENERAL COMMENTS	Main Comments: 1. Interesting findings with a large sample and a commendable response rate.2. While I believe it is a good idea to use Covid-19 to frame the discussion of the authors' findings (because it is true, as the authors point out in the Discussion section, that lockdown erased the safety net that some children rely on from in-person schooling, potentially worsening vulnerabilities), caution should be used in how results are reported and applied to the pandemic, as there is no way to know if the situation for these children were identical in 2020-21 (as data was collected 2016-2019). For example, in the Discussion, the authors state that, "In addition, the high prevalence of children with no access to the internet at home and no access to a garden or park mean that many children will have at times been unable to study or be physically active." (page 15, lines 26-30). Again, this refers to pre-pandemic numbers that could easily have changed before lockdown, and this is an overreach for cross-sectional results. There are certainly recommendations to be made for policy, as the authors aptly point out, and the future policy applications should be the focus of drawn conclusions.3. Reporting findings by ethnicity is always complex, especially in cross-sectional studies. The issue is fairly well dealt with here, although I'd expect some more in-depth statistical analysis to claim significance. Additionally, I would have liked to see some debate in the Discussion section around potential biases with regards to minority groups – is there social desirability bias in surveying? Are differences reported simply cultural differences? This I think may be a potential limitation and deserves discussion.4. Finally, as an overseas reviewer, the term "Gypsy" does not sit well with me, and I was thrown by the use of the term throughout the paper. Please refer to those classified as "Gypsy/Irish Travellers" as "Nomadic". Introduction Section: 1. Unless I am mistaken, BMJ Open does not limit the number of
--

	references authors may use (apologies if I am incorrect). Wellbeing is an enormous topic among children, so I was surprised to see only eight references represented in the Introduction; expansion is needed. For example, as specific vulnerabilities are the focus of the research, I would expect to see them introduced in the Introduction section with supporting literature – a paragraph or two should suffice. 2. Page 5 of Introduction, Lines 24-35 (beginning with “To provide ...”) belong at the end of the section, placed in line 45 before “The aim of this paper is...” Methods Section: 1. Figure 1 – this Figure reports results and should therefore be moved to that section. Also, please check n values in Figure 1, for example 15,641 + 2,133 equals 17,774, not 17,744. 2. Please state how missing ethnicity data was supplemented by data held by BiB. 3. Study protocol needs to be read in order to fully understand and repeat work but that is acceptable. Results Section: 1. Move Figure 1 here. Discussion: 1. See Main Comments above. Minor Comments: Some inconsistencies need to be addressed: 1. Watch use of tenses – check they are correct as I found some errors. 2. Always spell out numbers below 11 – this is only done intermittently – fix throughout document. 3. Never begin a sentence with a number (Results, page 9, lines 32 and 37). 4. Capitalization of vulnerability domains is inconsistent – pick lower case or caps and stick with it. 5. Use of parentheses when reporting percentages and 95% CIs is not consistent – please check document thoroughly, there are extras in some places. 6. Covid-19 or COVID-19 – pick one and stick with it.
--	--

REVIEWER	Chase, Elaine University College London
REVIEW RETURNED	03-Aug-2021

GENERAL COMMENTS	This is an important large-scale study providing descriptive data on the self-reported wellbeing of primary school-aged children in Bradford (a ‘deprived, multi-ethnic community’) across four domains (home, family and family relationships; material resources; friends and school; and subjective wellbeing). The statistical analysis appears to be sound (although I am not a statistician) and the analysis of findings are well presented in a clear and accessible style. I have a number of suggestions for strengthening the paper and hope these are helpful. 1) Whilst I recognise that the intention was to describe the prevalence of factors related to wellbeing among primary school children and that it is not possible to look at underlying causes, I think more could be done to engage with the findings in a more critical way and in relation to the broader literature. In particular, the discussion section tends to be highly repetitive of the findings
---

	section and more could be done to develop this further. 2) For example, the section on page 12 of the discussion, drawing out some of the gender differences, highlights particular vulnerabilities for boys compared to girls across all domains. The discussion then goes on to discuss the implications of this findings for girls rather than for boys (e.g by suggesting that poorer wellbeing outcomes for older, 'adolescent' girls could be prevented by somehow protecting the better outcomes at primary age). For me a major question jumps out in terms of what are the implications also for promoting boys' wellbeing at primary school level and beyond. 3) The section highlighting the differences in vulnerability according to ethnicity claims to help us better understand the 'nuances' in ethnic variations across 9 ethnic groupings. Again, whilst recognising the descriptive nature of the study, there still remains a question as to 'so what' in terms of how could these findings usefully be used to support the work of schools in promoting children's wellbeing and considering how ethnicity may be an important factor to engage with. 4) I was not sure of the relevance of the arguments about drawing out the distinction between the approach of the current study and the Adverse Childhood Experiences ACE) and the Children's Commissioner for England definition and measurement of extreme child vulnerabilities (p. 14). Would it not be more helpful methodologically to compare it to other studies of wellbeing (rather than adversity) such as other studies cited in the paper (By Bradshaw and colleagues and Ben Arie et al). 5) On page 14 the section on 'Implications for providers of services to children and policy makers' there is a sudden Segway into the context of the COVID-19 pandemic (when the study was conducted prior to the pandemic). I wondered whether there is a risk here in what has come to be termed the Covidisation of research. While of course the vulnerabilities described are likely to have been exacerbated during the time of the pandemic, is it not also important to consider their implications irrespective of the pandemic?. I would suggest starting this section with a discussion of the implications of the findings and then perhaps suggest that COVID-19 adds another layer of factors which need to be taken into account. 6) On page 15, there is suggestion that 'all schools work hard to create safe and supportive environments for children' – I think it is important to acknowledge that there is huge variation in how well schools do this. We know that the ethos of a school and attention to issues of wellbeing can be very different – so perhaps nuancing this statement will be really important. (It struck me that if it is possible to work with school level data, it would be fascinating to look at broader factors at the school level to see what explanatory value they have with respect to promoting or undermining children's wellbeing – although this of course is a separate study!) 7) The conclusions and implications for public health (page 15) are rather cursory and could go a bit further to really think about how commissioners in public health might use this data in strategic planning. I would also suggest that it makes more sense to group the implications sections (for provides of services to children and policymakers on page 14) and implications for public health (page 15) – since any response to these issues clearly requires an intersectoral approach.
--	---

VERSION 1 – AUTHOR RESPONSE

Reviewer: 1
Dr. Aimee Ward, Kent State University

Main Comments:

1. Interesting findings with a large sample and a commendable response rate.
Thank you for your positive comment.

2. While I believe it is a good idea to use Covid-19 to frame the discussion of the authors' findings (because it is true, as the authors point out in the Discussion section, that lockdown erased the safety net that some children rely on from in-person schooling, potentially worsening vulnerabilities), caution should be used in how results are reported and applied to the pandemic, as there is no way to know if the situation for these children were identical in 2020-21 (as data was collected 2016-2019). For example, in the Discussion, the authors state that, "In addition, the high prevalence of children with no access to the internet at home and no access to a garden or park mean that many children will have at times been unable to study or be physically active." (page 15, lines 26-30). Again, this refers to pre-pandemic numbers that could easily have changed before lockdown, and this is an overreach for cross-sectional results. There are certainly recommendations to be made for policy, as the authors aptly point out, and the future policy applications should be the focus of drawn conclusions. The data presented in this paper were indeed collected prior to the pandemic, and the manuscript submitted before any pandemic-related data were available for these children. We intended them to be the published descriptive baseline for research we conducted during the pandemic, which followed some of these children and assessed how their lives had been affected by Covid. The very long period under review has created a slight awkwardness in presenting this descriptive baseline distinct from later findings, some of which have now been published before this earlier paper. In the Discussion, we are now able to reference some of those new findings, which confirm our previous inferences, so our discussion of the impact of Covid on this sample is now less speculative than before. We believe our Discussion is now strengthened by being able to point to this additional evidence but that it is still valuable to set out a description of child wellbeing in the UK prior to the onset of Covid.

3. Reporting findings by ethnicity is always complex, especially in cross-sectional studies. The issue is fairly well dealt with here, although I'd expect some more in-depth statistical analysis to claim significance. Additionally, I would have liked to see some debate in the Discussion section around potential biases with regards to minority groups – is there social desirability bias in surveying? Are differences reported simply cultural differences? This I think may be a potential limitation and deserves discussion.

Ethnic variation in health and wellbeing has always been a focus of Born in Bradford, a city where more than half of new babies are born to parents belonging to ethnic minority communities. The current study is an expansion of the cohort to include whole classroom analyses of a large number of Bradford primary schools, which has enabled us to include the full range of ethnic groups present in the Bradford population of school children. With coverage of three quarters of primary schools and a response rate of 88% we do not believe that our findings are materially affected by selection or information bias with regards to ethnicity, and in fact debated whether statistical testing of ethnic variation was appropriate at all given our whole population approach. Nevertheless, analyses of ethnic differences with 95% confidence intervals are fully reported in Supplementary Table S4. In line with contemporary epidemiological methodological good practice, we have not estimated multivariable models that include ethnicity as an independent variable: VanderWeele TJ, Robinson WR. On the causal interpretation of race in regressions adjusting for confounding and mediating variables. *Epidemiology (Cambridge, Mass)*. 2014;25(4):473-84. This is to avoid the difficulties inherent in interpreting the ethnicity coefficient in regression models controlling for other variables. Additionally, in line with current UK social science practice, we believe that the description of ethnic differences is often appropriate in social policy oriented research and will be of interest to readers of this article:

Salway S, Barley R, Allmark P, Gerrish K, Higginbottom G, Ellison G. Ethnic diversity and inequality: ethical and scientific rigour in social research. York: Joseph Rowntree Foundation; 2011.

4. Finally, as an overseas reviewer, the term “Gypsy” does not sit well with me, and I was thrown by the use of the term throughout the paper. Please refer to those classified as “Gypsy/Irish Travellers” as “Nomadic”.

In the UK, Gypsy/Irish Traveller is an official classification used by the Office for National Statistics and is used in the UK Census, by schools and other institutions who collect data on ethnic identity. Gypsy and Irish Traveller identity is protected by the UK Equality Act of 2010. We are sorry that the reviewer is uncomfortable with this terminology but it has accepted usage in the UK for self-identification of ethnicity. It would not be appropriate for us to substitute this with a different term and we note that the category was agreed by the ONS in collaboration and engagement with Gypsy and Irish Traveller movements, local government and other agencies.

Introduction Section:

1. Unless I am mistaken, BMJ Open does not limit the number of references authors may use (apologies if I am incorrect). Wellbeing is an enormous topic among children, so I was surprised to see only eight references represented in the Introduction; expansion is needed. For example, as specific vulnerabilities are the focus of the research, I would expect to see them introduced in the Introduction section with supporting literature – a paragraph or two should suffice.

It is correct that BMJ Open does not limit number of references, although BMJ guidance across its journals suggests trying to limit to 20, but it does provide guidance on word count (~4000) and our manuscript is over 6000 words without counting words in figures, tables and supplementary materials. We wrote our Introduction in line with BMJ guidance for authors across its journals that requests “a succinct introduction that focuses — in no more than three paragraphs — on the background to the research question (<https://www.bmj.com/sites/default/files/attachments/resources/2018/05/BMJ-InstructionsForAuthors-2018.pdf>). Wellbeing is indeed a vast topic but we have focused our introduction on the lack of available data on contemporary child wellbeing in the UK rather than on why wellbeing or vulnerabilities are of interest per se or should be measured.

2. Page 5 of Introduction, Lines 24-35 (beginning with “To provide ...”) belong at the end of the section, placed in line 45 before “The aim of this paper is...”

Thank you, this change has been made.

Methods Section:

1. Figure 1 – this Figure reports results and should therefore be moved to that section. Also, please check n values in Figure 1, for example $15,641 + 2,133$ equals 17,774, not 17,744.

Figure 1 has been moved to the Results and we have corrected the typo

2. Please state how missing ethnicity data was supplemented by data held by BiB.

This information has been added to page 8.

3. Study protocol needs to be read in order to fully understand and repeat work but that is acceptable.

Thank you.

Results Section:

1. Move Figure 1 here.

Done

Discussion:

1. See Main Comments above.

See responses above

Minor Comments:

Some inconsistencies need to be addressed:

1. Watch use of tenses – check they are correct as I found some errors.

Thank you. We have decided to use past tense for reporting of prevalence (eg, 10% of children had no vulnerabilities) and present tense for description of our actions (eg, we present means and confidence intervals in Table X). We have checked for consistency throughout.

2. Always spell out numbers below 11 – this is only done intermittently – fix throughout document. We checked BMJ house style (see link above), which asks that numbers under 10, rather than 11, are spelt out, except for measurements with a unit (eg 10%) or age (age 7-10 years), or when in a list with other numbers. We have made changes throughout in accordance.

3. Never begin a sentence with a number (Results, page 9, lines 32 and 37).

BMJ house style is that “Numbers over 10 do not need spelling out at the start of sentences”

4. Capitalization of vulnerability domains is inconsistent – pick lower case or caps and stick with it.

Thank you. We have decided to capitalise the domain labels throughout.

5. Use of parentheses when reporting percentages and 95% CIs is not consistent – please check document thoroughly, there are extras in some places.

Thank you, we have checked and corrected throughout

6. Covid-19 or COVID-19 – pick one and stick with it.

BMJ uses Covid-19 so we have corrected throughout, except in references which have used COVID-19, thank you.

Reviewer: 2

Dr. Elaine Chase, University College London

Comments to the Author:

This is an important large-scale study providing descriptive data on the self-reported wellbeing of primary school-aged children in Bradford (a ‘deprived, multi-ethnic community’) across four domains (home, family and family relationships; material resources; friends and school; and subjective wellbeing). The statistical analysis appears to be sound (although I am not a statistician) and the analysis of findings are well presented in a clear and accessible style. I have a number of suggestions for strengthening the paper and hope these are helpful.

Thank you for these positive comments and suggestions

1) Whilst I recognise that the intention was to describe the prevalence of factors related to wellbeing among primary school children and that it is not possible to look at underlying causes, I think more could be done to engage with the findings in a more critical way and in relation to the broader literature. In particular, the discussion section tends to be highly repetitive of the findings section and more could be done to develop this further.

This is a helpful point, and we have added several paragraphs to the Discussion relating our findings to the literature and describing what can be done to improve wellbeing at the start of the section on Implications for providers of services to children and policymakers (p 14-15).

2) For example, the section on page 12 of the discussion, drawing out some of the gender differences, highlights particular vulnerabilities for boys compared to girls across all domains. The discussion then goes on to discuss the implications of this findings for girls rather than for boys (e.g by suggesting that poorer wellbeing outcomes for older, 'adolescent' girls could be prevented by somehow protecting the better outcomes at primary age). For me a major question jumps out in terms of what are the implications also for promoting boys' wellbeing at primary school level and beyond.

This is an important point. We have included some discussion of promoting boys' wellbeing. Importantly, we will be able to review whether this gender difference persists after children have started secondary school, and have noted this in the discussion (p 12-13).

3) The section highlighting the differences in vulnerability according to ethnicity claims to help us better understand the 'nuances' in ethnic variations across 9 ethnic groupings. Again, whilst recognising the descriptive nature of the study, there still remains a question as to 'so what' in terms of how could these findings usefully be used to support the work of schools in promoting children's wellbeing and considering how ethnicity may be an important factor to engage with.

In our discussion on what can be done (p14-15) we have included information on what could be done to address inequalities by ethnicity.

4) I was not sure of the relevance of the arguments about drawing out the distinction between the approach of the current study and the Adverse Childhood Experiences ACE) and the Children's Commissioner for England definition and measurement of extreme child vulnerabilities (p. 14). Would it not be more helpful methodologically to compare it to other studies of wellbeing (rather than adversity) such as other studies cited in the paper (By Bradshaw and colleagues and Ben Arieh et al).

We had initially felt that some readers might wonder why we were not simply taking an ACE approach to vulnerability and that we should describe the difference in our approach, however we take the Reviewers point and have modified the paragraph to simply say that we wished to (a) measure vulnerability rather than adversity, (b) not just extreme vulnerability but a broader swathe of children's experiences and (c) wellbeing itself rather than social indicators of wellbeing, including well-becoming.

5) On page 14 the section on 'Implications for providers of services to children and policy makers' there is a sudden segue into the context of the COVID-19 pandemic (when the study was conducted prior to the pandemic). I wondered whether there is a risk here in what has come to be termed the Covidisation of research. While of course the vulnerabilities described are likely to have been exacerbated during the time of the pandemic, is it not also important to consider their implications irrespective of the pandemic?. I would suggest starting this section with a discussion of the implications of the findings and then perhaps suggest that COVID-19 adds another layer of factors which need to be taken into account.

In the Implications section, we have added some paragraphs on general implications for services and national/local policy, prior to discussion of the role of Covid-19 (p 14-15).

Also, please see response to Reviewer 1 and the revision of our Discussion in light of our follow-up of these children through the pandemic.

6) On page 15, there is suggestion that 'all schools work hard to create safe and supportive environments for children' – I think it is important to acknowledge that there is huge variation in how well schools do this. We know that the ethos of a school and attention to issues of wellbeing can be very different – so perhaps nuancing this statement will be really important. (It struck me that if it is possible to work with school level data, it would be fascinating to look at broader factors at the school level to see what explanatory value they have with respect to promoting or undermining children's

wellbeing – although this of course is a separate study!)

This is a good point and we have nuanced this statement. Although all schools say they don't tolerate bullying, for example, clearly, from our findings, they don't manage to create such safe environments – three needs to be greater focus on this at school level, and in resourcing for schools (p14-15).

7) The conclusions and implications for public health (page 15) are rather cursory and could go a bit further to really think about how commissioners in public health might use this data in strategic planning. I would also suggest that it makes more sense to group the implications sections (for providers of services to children and policymakers on page 14) and implications for public health (page 15) – since any response to these issues clearly requires an intersectoral approach.

We have added some sentences providing further implications for public health – both in the implications section in the discussion and in the conclusion. We completely agree – our findings demonstrate the importance of a multi-sectoral approach with integrated, intersectoral pathways of support.

VERSION 2 – REVIEW

REVIEWER	Ward, Aimee Kent State University, Geography
REVIEW RETURNED	05-Jan-2022

GENERAL COMMENTS	Review checklist: #2: Abstract is much too long #8: Number of references used are insufficient for the topic See below document for additional comments to authors: Manuscript ID bmjopen-2021-049416.R1 Dr Aimee L Ward comments The implications for policy that were included in the discussion on the suggestion of Reviewer 2 are a great addition and have strengthened that section. The reordering of the introduction, and movement of Figure 1 to the results section, are also an improvement. However, I still have a few concerns. Overall comments: 1. More and more, scientific writers are being encouraged to use the term “sex” instead of “gender”, because gender is a social construct. Throughout the paper, please consider replacing “gender” with “sex”. 2. Regarding the following statements from the introduction: “Wellbeing declines as children and young people grow older” (line 17), and “The impact of ethnicity on child wellbeing at any age is unclear and most previous UK studies have been unable to examine wellbeing across a wide range of ethnicities.” (lines 22-24): These are HUGE statements, and in this manuscript, both supported by the same single reference, a UK government report. In my previous review, I requested that more references be included (and possible expansion of Intro be considered) since child wellbeing is such a vast topic. This suggestion was deemed inappropriate by the authors, and the reason given was that the introduction needed to be succinct, as follows: “It is correct that BMJ Open does not limit
--

number of references, although BMJ guidance across its journals suggests trying to limit to 20, but it does provide guidance on word count (~4000) and our manuscript is over 6000 words without counting words in figures, tables and supplementary materials. We wrote our Introduction in line with BMJ guidance for authors across its journals that requests a succinct introduction that focuses — in no more than three paragraphs — on the background to the research question. Wellbeing is indeed a vast topic but we have focused our introduction on the lack of available data on contemporary child wellbeing in the UK”.

I note that the paper’s abstract currently comes in at over 600 words, while the BMJ Open guidelines limit abstracts to 300 words – so, surely the introduction can provide more context and words can be cut from the abstract, if that is indeed a concern. I am not suggesting that the introduction be expanded by a lot; I am simply advising that the child wellbeing literature be given the thought and consideration that it deserves, by adding a few sentences to provide a more thorough and international context. I am concerned that broad statements are being made without proper referencing, especially considering there are many seminal works in this field. The entire introduction includes only eight references, which is simply insufficient. An introduction is meant to not only provide a succinct summary of the importance of the research at hand, but to also provide a brief overview that places that research in the broader context of the literature as a whole. As it stands currently, it does not accomplish this.

3. In my first review, I stated some concern about reporting findings by ethnicity, and discussion of any potential biases, especially due to the cross-sectional nature of the research. In their response to my comment, the authors stated, “With coverage of three quarters of primary schools and a response rate of 88% we do not believe that our findings are materially affected by selection or information bias with regards to

ethnicity, and in fact debated whether statistical testing of ethnic variation was appropriate at all given our whole population approach.... In line with contemporary epidemiological methodological good practice, we have not estimated multivariable models that include ethnicity as an independent variable:

VanderWeele TJ, Robinson WR. On the causal interpretation of race in regressions adjusting for confounding and mediating variables. *Epidemiology (Cambridge, Mass)*. 2014;25(4):473-84. This is to avoid the difficulties inherent in interpreting the ethnicity coefficient in regression

models controlling for other variables. Additionally, in line with current UK social science practice, we believe that the description of ethnic differences is often appropriate in social policy oriented research and will be of interest to readers of this article: Salway S, Barley R, Allmark P, Gerrish K, Higginbottom G, Ellison G. *Ethnic diversity and inequality: ethical and scientific rigour in social research*. York: Joseph Rowntree Foundation; 2011.”

This response from the authors is a beautiful paragraph that resolves my concern completely, and includes important references that support the authors’ decisions; thus, this justification should be included in the manuscript’s discussion in its entirety.

4. The first paragraph of the discussion is simply a rehashing of results and should be deleted from the section; any new information in this paragraph should be moved to the results section.

	5. In the discussion, I find it interesting that there are no references to other highprofile and world-renowned birth cohort studies outside of the UK, such as the Dunedin Multidisciplinary Health and Development Study that has been going on for half a century. (https://dunedinstudy.otago.ac.nz/) – this seems like a huge oversight. I would suggest the authors compare their findings with other cohorts outside the UK. I understand that the focus of the article is on child wellbeing in the UK, but the purpose of peer-reviewed publications is to place work in context with existing research.
REVIEWER	Chase, Elaine University College London
REVIEW RETURNED	01-Jan-2022
GENERAL COMMENTS	This is clearly written and important paper - I think the authors have done a great job at responding to earlier peer review suggestions.

VERSION 2 – AUTHOR RESPONSE

Reviewer 2:

Thank you for your kind words about our revised manuscript and the importance of our paper

Reviewer 1:

We are glad that the reviewer liked the changes we made to consider policy implications and to re-order the introduction and move Figure 1. In response to her further comments:

1. We have replaced 'gender' with 'sex' throughout
2. We agree, of course, that the literature on child wellbeing is vast. The focus of our introduction in has been to situate our descriptive study of child wellbeing in a single British city within the UK context, where we have a dearth of empirical data for this age group. Our reference for what the reviewer considers to be 'huge statements' was a government-commissioned report that provided an excellent overview of the peer-reviewed literature in this area - which pointed to the lack of evidence on wellbeing by ethnicity and summarised the evidence on the decline of wellbeing with age. We do not agree with the reviewer that the child wellbeing literature has not been "given the thought and consideration it deserves" as we are citing an 87-page report that includes the most up-to-date national data on child wellbeing for the UK and itself cites an extensive literature, nor do we agree that by only citing eight references in the introduction we are giving readers insufficient evidence to understand that there is a research gap that we are addressing with our study. The reviewer asks that we include a few sentences that place our study in an international, comparative context; we did that in the first sentence of our paper, with reference to the seminal Unicef study of child wellbeing in rich countries, which was the first truly comparable analysis of a large number of developed countries. We feel that our research question is well justified and contextualised by the literature we cite.
3. We have taken the reviewers suggestion and included the text from our previous response in the revised manuscript on page 13.
4. The reviewer asks that we delete the first paragraph of the Discussion section Statement of Principal Findings. We have shortened this considerably.

5. Again, the reviewer is asking us to compare our findings to those of other cohort studies around the world, eg the Dunedin study in New Zealand. Again, we respectfully disagree that this is an oversight. Our study is a descriptive snapshot of children in a single city at a point in time (the fact that many of our children belong to a longitudinal cohort is not of relevance to this paper) - our aim was to provide evidence on wellbeing among primary school aged children from a variety of UK ethnicities. We are not attempting to address the question of whether or not children in Bradford have better or worse wellbeing than children in other countries and while we agree with the reviewer that "the purpose of peer-reviewed publications is is to place work in context feel that we have done this in a way that will make sense to readers.